# Understanding the Molecular Conformation and Viscoelasticity of Low Sol-Gel Transition Temperature Gelatin Methacryloyl Suspensions

**DOI:** 10.3390/ijms24087489

**Published:** 2023-04-19

**Authors:** Cristina Padilla, Franck Quero, Marzena Pępczyńska, Paulo Díaz-Calderon, Juan Pablo Acevedo, Nicholas Byres, Jonny J. Blaker, William MacNaughtan, Huw E. L. Williams, Javier Enrione

**Affiliations:** 1Programa de Doctorado en Biomedicina, Facultad de Medicina, Universidad de los Andes, Santiago 7620086, Chile; 2Centro de Investigación e Innovación Biomédica (CIIB), Universidad de los Andes, Santiago 7620086, Chile; 3Biopolymer Research and Engineering Laboratory (BIOPREL), Escuela de Nutrición y Dietética, Facultad de Medicina, Universidad de Los Andes, Santiago 7620086, Chile; 4IMPACT, Center of Interventional Medicine for Precision and Advanced Cellular Therapy, Universidad de los Andes, Santiago 7620086, Chile; 5Laboratorio de Nanocelulosa y Biomateriales, Departamento de Ingeniería Química, Biotecnología y Materiales, Facultad de Ciencias Físicas y Matemáticas, Universidad de Chile, Santiago 8370456, Chile; 6Department of Materials and Henry Royce Institute, The University of Manchester, Manchester M13 9PL, UK; 7Department of Biomaterials, Institute of Clinical Dentistry, University of Oslo, 0317 Oslo, Norway; 8Division of Food, Nutrition and Dietetics, School of Biosciences, Sutton Bonington Campus, University of Nottingham, Loughborough LE12 5RD, UK; 9Centre for Biomedical Sciences, University Park, University of Nottingham, Nottingham NR7 2RD, UK

**Keywords:** gelatin, GelMA, sol-gel transition temperature, molecular configuration

## Abstract

For biomedical applications, gelatin is usually modified with methacryloyl groups to obtain gelatin methacryloyl (GelMA), which can be crosslinked by a radical reaction induced by low wavelength light to form mechanically stable hydrogels. The potential of GelMA hydrogels for tissue engineering has been well established, however, one of the main disadvantages of mammalian-origin gelatins is that their sol-gel transitions are close to room temperature, resulting in significant variations in viscosity that can be a problem for biofabrication applications. For these applications, cold-water fish-derived gelatins, such as salmon gelatin, are a good alternative due to their lower viscosity, viscoelastic and mechanical properties, as well as lower sol-gel transition temperatures, when compared with mammalian gelatins. However, information regarding GelMA (with special focus on salmon GelMA as a model for cold-water species) molecular conformation and the effect of pH prior to crosslinking, which is key for fabrication purposes since it will determine final hydrogel’s structure, remains scarce. The aim of this work is to characterize salmon gelatin (SGel) and salmon methacryloyl gelatin (SGelMA) molecular configuration at two different acidic pHs (3.6 and 4.8) and to compare them to commercial porcine gelatin (PGel) and methacryloyl porcine gelatin (PGelMA), usually used for biomedical applications. Specifically, we evaluated gelatin and GelMA samples’ molecular weight, isoelectric point (IEP), their molecular configuration by circular dichroism (CD), and determined their rheological and thermophysical properties. Results showed that functionalization affected gelatin molecular weight and IEP. Additionally, functionalization and pH affected gelatin molecular structure and rheological and thermal properties. Interestingly, the SGel and SGelMA molecular structure was more sensitive to pH changes, showing differences in gelation temperatures and triple helix formation than PGelMA. This work suggests that SGelMA presents high tunability as a biomaterial for biofabrication, highlighting the importance of a proper GelMA molecular configuration characterization prior to hydrogel fabrication.

## 1. Introduction

Gelatin is a hydrocolloid of animal origin that has been used for many years in the food and pharmaceutical industries, mainly as a jellifying and thickening agent [1,2,3]. Gelatin is obtained through the partial hydrolysis of collagen fibers extracted from skin, cartilage, bones and/or hair of animals [4] and is composed of a mixture of different polymeric structures: α-chains, β-chains (two α-chains) and γ-chains (three α-chains) [4,5]. These α-chains present a polyproline II conformation, which requires a repetitive peptide Glycine-X-Y sequence, with the iminoacids proline (Pro) and hydroxyproline (Hyp) most frequently located in the X and Y position, respectively [6,7]. This particular molecular configuration plays an important role in the stability of the helical structures associated with the partial renaturation of gelatin [6]. Gelatin extraction protocols include the use of acid or alkaline chemicals for hydrolysis at temperatures between 50 and 80 °C [8]. This has been found to significantly affect the molecular weight and isoelectric point, producing two well-defined types of gelatins: type A (acid hydrolysis), with an isoelectric point at pH 7–9, and type B (alkaline hydrolysis), with an isoelectric point near pH 5–6 [4,9,10].

In recent years, gelatin has also been extensively used for studies in tissue engineering and in medical devices, due to its viscoelasticity, biocompatibility, biodegradability, and the presence of biochemical cues on its structure (RGD peptides) [11,12]. For biomedical applications, gelatin covalent crosslinking is often necessary to stabilize such hydrogels as well as improve their mechanical properties [13]. The most used strategies include gelatin chemical modification with photocroslinkable groups such as acrylate, methacryloyl and norbornene [14]. The most common strategy though, corresponds to gelatin modification with methacryloyl groups to obtain gelatin methacryloyl (GelMA), followed by radical-induced crosslinking of these groups by the addition of a photo-initiator and light induction using ultraviolet or visible light [15,16,17]. By varying the amount of methacrylic anhydride added during GelMA production, the degree of substitution (or functionalization) can be adjusted [17]. This method can produce hydrogels that are stable under physiological temperatures with tunable structural and mechanical properties, which can be achieved by modifying: the degree of substitution (DS) of GelMA, the concentration of GelMA in the suspension, the time of exposure, intensity, and distance of the light source, the concentration of the photo-initiator, and temperature before crosslinking [13,15,17,18,19,20,21]. This structural and mechanical control is very important for TE since it determines degradability, cell infiltration to the scaffold, proliferation, and cell differentiation to specific tissues [22,23,24]. This has given rise to bovine- and porcine-derived GelMA being widely studied for biomedical applications [13,16,25,26], showing high biocompatibility, scaffold integration in vivo, as well as its contribution to cell differentiation, regeneration, and vascularization in GelMA-based scaffolds aimed for skin TE [27,28,29,30,31,32]. This structural and mechanical tunability of GelMA hydrogels has also been used for the production of drug delivery carriers, showing an efficient delivery of different active molecules such as growth factors and anticancer drugs [33,34]. However, in mammalian-origin gelatins, sol-gel transitions are close to room temperature and can result in significant variations in viscosity [35]. This can be a problem/limitation for biofabrication systems such as bioprinting and microfluidics because mammalian-derived GelMA commonly requires heating to lower its viscosity, hence a uniform temperature control over the entire fabrication process is needed to maintain the viscosity of materials, increasing the cost and complexity of the fabrication process, since if this is not controlled, it could eventually affect the final hydrogel properties [35,36]. For these applications, cold-water fish-derived gelatin is a good alternative due to its marked differences in terms of rheological, viscoelastic, and mechanical properties and lower sol-gel transition temperature when compared with mammalian gelatins [10,37,38,39,40,41,42]. For example, gelatin obtained from salmon skin has shown low viscosity, maintaining its liquid form even at temperatures as low as 5 °C [43]. This characteristic has been explained by a lower content of Pro and Hyp amino acids, and therefore a reduced number of triple helices at room temperature in comparison with its mammalian gelatin counterparts due to evolutionary adaptation to low-temperature environments [10,38,44]. Similar to mammalian gelatin, salmon gelatin can also be chemically modified when aiming at similar technological applications, although it has not been widely studied. Work by Yoon et al. proposed the production of methacryloyl cold-water fish gelatin (undefined) as a strategy to produce more versatile GelMA hydrogels for tissue engineering applications, which exhibit lower mechanical strength, higher water swelling degree and degradation rates, and similar biocompatibility in comparison to porcine GelMA. However, the authors did not provide information regarding its specific gelatin source (i.e., species and tissue), amino acid content or viscoelasticity [45]. Young et al. evaluated the differences presented by GelMA hydrogels from porcine, bovine, and cold-water fish origins (undefined) concerning their viscoelastic properties and hydrogel density, highlighting the importance of choosing the appropriate gelatin source to produce GelMA hydrogels [21]. Furthermore, Zaupa et al. showed that GelMA hydrogels obtained with salmon and bovine origins displayed similar compression modulus at 40 °C (over gelation temperature for both gelatin types), but different pore sizes at the same degree of functionalization, which increased the cell-remodeling rate, suggesting that salmon GelMA hydrogels present higher molecular mobility [36]. This particular characteristic of GelMA with a salmon origin has allowed its use for the development of a bioink formulation for its use in high-resolution 3D printing systems such as polyjet 3D printing, showing that this formulation achieved a high viability (∼80%) and proliferation of co-printed cells, while demonstrating in vivo the immune tolerance of printed structures [46].

Due to these findings and its wide temperature processing window without significant changes in viscosity at room temperature, salmon gelatin possesses significant potential for the development of novel scaffold structures and medical devices, where fine control of shape geometry and tuned physical and mechanical properties are paramount. These properties have been shown to be controlled by several extraction variables including pH and time [8] and/or during functionalization, and hydrogel fabrication by varying different process parameters aforementioned. Additionally, pH was shown to modulate the structuring of the gelatin polymer chain conformation in aqueous suspensions, affecting different gelatin properties including its gelling and melting temperatures [47]. Another advantage is that salmon-derived gelatin is a highly abundant byproduct and can be of consistent quality due to salmon farming following highly standardized protocols. Salmon gelatin provides a lower risk of pathogen vector transmittance in the form of prions [41,48,49] and does not have many of the religious restrictions that porcine and bovine gelatin products have [3,50].

Although the potential of mammalian GelMA hydrogels in tissue engineering and biofabrication has been established, information regarding GelMA (with special focus on salmon GelMA as a model for cold-water species) molecular conformation and the effect of pH prior to crosslinking, which is key for fabrication purposes since it will affect the final hydrogel’s structure, remains scarce. Consequently, the aim of this work is to characterize salmon gelatin (SGel) and salmon methacryloyl gelatin (SGelMA) at two different acidic pHs (3.6 and 4.8), since both production processes (gelatin extraction and methacrylation) occur under these conditions. We assessed their rheological and thermophysical properties as well as their molecular configuration compared to commercial porcine gelatin (PGel) and methacryloyl porcine gelatin (PGelMA). We hope to provide new key information and to highlight the importance of a proper GelMA molecular configuration characterization, which could contribute to the design of non-mammalian gelatin-derived biomaterials with tailored properties for a wider range of biomedical applications.

## 2. Results

### 2.1. Amino Acid Composition of Salmon and Porcine Gelatins

The quantitative analysis of the main amino acids present in the gelatins shows that salmon gelatin is constituted with a lower concentration of Pro and Hyp than porcine gelatin, as shown in Table 1. These results are consistent with the previous literature that compares the amino acid composition of cold-water fish and mammalian gelatin [38,51,52]. Lysine is also an amino acid of special interest since it is the main amino acid that is functionalized during the methacrylation process. Results show that both gelatins present very similar lysine contents, with similar degrees of functionalization expected under the same reaction conditions. 

### 2.2. Determination of Degree of Functionalization in GelMA

The colorimetric OPA method and ^1^H NMR were used to quantify the α-amino groups from the NH_2_-terminal of polypeptides and ε-amino groups from lysine, the main amino acid that participates in the functionalization of gelatin [15,53]. The degree of functionalization (DF) determined from both methods is reported in Table 2, which resulted in DF values over 90% in the case of SGelMA and over 83% for PGelMA, suggesting a high degree of functionalization for both gelatin sources under the evaluated conditions. This is expected since methacrylic anhydride was added in excess compared to the concentration of lysine present in the gelatin to ensure the full functionalization of this amino acid [18,19]. It is important to comment that ε-amino groups from lysine (methacrylamide groups) are not the only functionalized groups present in GelMA, since methacrylate groups, formed by the reaction of methacrylic groups with hydroxyl groups from serine, threonine, hydroxyproline, and hydroxylysine, are also present. However, it has been reported that under similar reaction conditions, the percentage of methacrylate groups in GelMA is less than 10% [54]. ^1^H NMR spectra for the different samples are depicted in the Appendix A.

### 2.3. Molecular Weight and ζ-Potential of Salmon and Porcine Gelatins

The molecular weight distributions for salmon and porcine gelatin assessed by SDS-PAGE are shown in Figure 1. Both gelatin sources display different molecular weight distributions but with α-chains present as major constituents. For salmon gelatin, alpha chains are within the range of 100–110 kDa. Other strong bands can be seen in the range of 70–75 kDa and at 50 kDa, probably due to hydrolytic effects that may occur upon extracting gelatin under acidic conditions [8,51,55]. In the case of porcine gelatin, alpha chain bands are located within slightly higher-range Mw values of 110–125 kDa, in agreement with previous reports [51]. The dissimilarity in molecular weight distribution has been attributed to differences in the aminoacidic sequence between different gelatin sources [50].

Regarding the methacryloyl gelatins, the results differ from the non-functionalized gelatins showing a less defined molecular weight profile. This result was further explored by increasing the acrylamide concentration of the electrophoresis gels, using different denaturation conditions and different buffers to facilitate the migration of the sample in the acrylamide gel, where no defined bands positioned at 125 kDa or lower could be identified (Figure 1).

This suggests that gelatin functionalization with methacrylic anhydride may be affecting gelatin molecular weight. In fact, a similar result was recently reported for porcine and bovine GelMA [56], although the authors did not discuss the lack of defined gelatin bands or molecular weight observed for their GelMA samples. However, the main interaction mechanism of Coomassie blue dye (used for staining proteins in the SDS-PAGE gel as indicated in Section 4.6.1) is through electrostatic interactions with positively charged amino acids [57], mainly absent from GelMA samples, which could also explain the lesser-defined Mw bands compared to the non-functionalized gelatin samples observed in Figure 1.

The differences in molecular weight due to gelatin functionalization were also suggested by the capillary viscometry (Appendix A). These data show a significant reduction in both reduced and inherent viscosity for GelMA compared to non-functionalized gelatin. A double extrapolation of these data allowed for the determination of the intrinsic viscosity for all sample types (Table 2), showing that GelMA samples have lower intrinsic viscosity values compared to the non-functionalized gelatins. The average molecular weight (Mw) values obtained from the intrinsic viscosity for PGel and SGel were higher (87 kDa and 73 kDa, respectively) compared with PGelMA (21 kDa) and SGelMA (23 kDa) (Table 2). The Mw values obtained for SGel were slightly lower than those previously reported in the literature [44,52,55]. This difference might relate to variations in the processing conditions used during gelatin extraction. Regarding the functionalized gelatins, the lower Mw values obtained are consistent with the results observed by SDS–PAGE and support the hypothesis that gelatin’s functionalization with methacrylic anhydride can affect gelatin’s molecular weight.

Previous studies using gel permeation chromatography have suggested that methacrylation decreases the molecular weight of GelMA [58]. Other authors argued that delayed elution times with an increased functionalization degree could be due to an alteration of the samples’ chain conformation in an aqueous solution [59]. To complement our analysis, SGel and SGelMA samples were analyzed by MALDI-TOF MS (Figure 2). Even though it was difficult to obtain clear signals from these samples, SGel seems to show higher molecular weight signals than SGelMA (the arrowheads in Figure 2). Importantly, similar molecular weight signals could be identified, which agreed with the SDS-PAGE results obtained for SGel (95 kDa, 70 kDa, 47 kDa, 35 kDa). These results suggested that salmon gelatin methacryloyl functionalization generates the hydrolysis of the protein, since the higher Mw signals were not detected by this technique.

The ζ-potential was quantified at various pH values ranging from 4 to 11 (Appendix A). An increase in the pH value of the different gelatin suspensions led to a decrease in ζ-potential values. Since both PGel and SGel are type A gelatins, isoelectric point (IEP) values in the basic pH range were expected. This was confirmed with values of 9.4 and 9.2 obtained for PGel and SGel, respectively (Table 2). Functionalization significantly affected ζ-potential, decreasing its values, and consequently decreasing its IEP to ~5 for both functionalized gelatins (Table 2). This is due to a decrease in the number of amine groups (positive charges) along the gelatin’s backbone structure, most likely due to functionalization. The values obtained for SGel and SGelMA agree with the results recently reported [36]. Another study described a similar effect on IEP values upon gelatin functionalization, suggesting that independent of the degree of functionalization, bovine- and porcine-derived GelMA exhibited an IEP < 5 [59].

After highlighting the differences in molecular structure between salmon and porcine gelatins and their functionalized counterparts, it is important to confirm if these differences would be mainly related to the pH of the samples or due to major changes in their molecular configuration.

### 2.4. Secondary Structure Molecular Configuration

Collagen molecular configuration has been widely studied by optical techniques such as optical rotation and circular dichroism (CD) [10,60]. The CD of collagen has shown a polyproline II-like secondary structure, with a positive band at around 220 nm and a large negative band in the region of 190–200 nm [60,61,62]. The intensity of the ~220 nm band has been attributed to the right-handed triple helix of collagen [63,64] and the intensity of the 200 nm band to random polypeptide chains [65]. All gelatin solutions studied at 40 °C show the characteristic spectral signature of collagen-like coiled helices with a negative trough at 200 nm, suggesting a random polypeptide conformation (Figure 3). The CD spectra for the cooled PGel samples at 4 °C at pH 4.8 show an increase in ellipticity at the 220 nm peak and a decrease at the 200 nm peak, which would indicate a higher content of triple helices due to lower temperatures (Figure 3b). SGel also shows a similar trend, though with a reduced thermal dependency compared to PGel. In fact, even though the 220 nm band for SGel at pH 4.8 is higher at 4 °C, it still shows negative values, suggesting less triple helix formation (Figure 3a). This is probably due to the lower Pro and Hyp content, as described in previous sections. Both gelatins present a slight increase in the maximum values of ellipticity at 220 nm, when the pH value was increased from 3.6 to 4.8, as it approaches their IEP value (Figure 3a,b). These findings are in accordance with reported results for type B bovine gelatin [64], where it was suggested that the triple helical content of a gelatin suspension shows the highest value close to the IEP. The latter can be explained in terms of a reduced interaction between the polymer chains and their polar surroundings, promoting a-chain interactions.

After functionalization, both gelatin structures were affected, and we noted that the overall magnitude of ellipticity values for the methacryloyl samples are slightly smaller, suggesting less triple helix formation (Figure 3). These results are in accordance with the literature that show a decrease in triple helix formation upon cooling after functionalization [20,66]. When comparing PGel with its methacryloyl counterpart, triple helix formation can still be seen when cooled at 4 °C at both pH values; however, there is a higher ellipticity when pH decreases to 3.6 (Figure 3b,d). In the case of SGelMA, the thermally induced triple helix formation upon cooling from 40 to 4 °C is hampered, evidenced by a reduction in intensity of the 220 nm band in comparison with SGel (Figure 3a,c). Small differences in the peak magnitude were observed between both pH values tested. This result therefore highlights the nature of the polymer chain stabilization for SGelMA, where pH seems to have no effect at the temperatures studied. This is not the case for PGelMA, where triple helix formation can still be seen upon cooling at both pH values. Similar CD results for SGel and SGelMA have been reported, showing a consistent trend for both types of gelatin and methacryloyl counterparts [36], although the effect of different pH was not evaluated in this case.

### 2.5. Rheological Characterization

The apparent viscosities of gelatin and GelMA suspensions at the two pHs (3.6 and 4.8) were evaluated as a function of temperature and shear rate. The viscosity–temperature ramps upon cooling for SGel and PGel can be observed in Figure 4a,b, respectively. As expected, PGel showed significantly higher viscosity (especially at lower temperatures) compared to SGel at both pH values. Differences in viscosity at 40 °C were not significant between PGel and SGel (~17 cP and ~11 cP, respectively); however, as the samples started to cool, the difference increased (~340 cP and ~19 cP for PGel and SGel, respectively, at 20 °C). As aforementioned, differences in viscosity at higher temperatures have been associated with differences in gelatin molecular weight. Upon cooling, a re-association of α-chains occurs, forming triple helices (gelation), which can be evidenced by a rapid increase in viscosity that occurs at different temperatures for PGel (26 to 28 °C) and SGel (6 to 10 °C) (Figure 4). This difference has been attributed to variations in amino acid composition between mammals and cold-water adapted marine species [10,48,67].

Functionalization by methacryloyl groups produced a significant decrease in viscosity for both types of gelatin (PGelMA and SGelMA). In the case of mammalian gelatin, this effect has been attributed to a reduction in the intermolecular forces among inner polymer chains due to methacryloyl functionalization [13,18,19,20,59]. However, findings from molecular weight studies suggest that the functionalization could also affect the molecular weight of gelatin, which could also be a factor contributing to this decrease in viscosity. Differences in viscosity were also observed at both pH values studied, at temperatures below the marked increase in viscosity (arrows), which was more evident in SGel and SGelMA samples. The increased viscosity upon cooling observed for all gelatin samples suggests that the triple helical formation kinetics might be favored at pH 4.8. This is consistent with CD data, indicating a higher number of triple helices for the samples at this pH value.

Within the shear rate deformation range tested, SGel and SGelMA suspensions exhibit a mostly Newtonian behavior at 20 °C (Appendix A). This feature has been identified as a viscoelastic requirement for novel applications such as inkjet 3D-based bioprinting [36,68]. The apparent viscosity of SGelMA at pH 4.8 was higher than at pH 3.6. PGel and PGelMA could not be evaluated at these conditions (10% *w*/*v*) due to gelling at 20 °C.

Figure 5a shows G′ and G″ values of SGel upon cooling from 40 °C to −5 °C. The data show clearly that the gelation point, defined as the crossing between G′ and G″ curves, is reduced by changing the pH from 4.8 to 3.6 for all the gelatin suspensions. Indeed, as it is indicated in Table 3, SGel’s gelation point shifted from 7.0 to 4.5 °C and SGelMA from 2.4 to −0.4 °C. PGel and PGelMA behaved similarly, but the shifts were smaller (from 25.3 to 24.2 °C and from 18.5 to 17.0 °C, respectively) (Figure 5b and Table 3). The gelling temperatures reported in Table 3 are in agreement with the viscosity values reported in Figure 4, which were higher at the lower temperature range for the samples at pH 4.8 compared to pH 3.6. This also agrees with the CD results, where a lower thermal sensitivity at a wide temperature range was observed for SGel and no thermal dependency was observed for SGelMA, since their gelling temperatures were lower compared to both PGel and PGelMA. This, however, might have not been completely achieved at 4 °C during the CD measurements.

Overall, the CD and rheological analysis suggests that for gelatins and GelMA from different origins, a decrease in the pH value from 4.8 to 3.6 affects their molecular configuration, decreasing their capability to form triple helices while cooling. Specifically, gelatin and GelMA of salmon origin are more sensitive to this pH decrease, showing an overall decrease of >2.5 °C in their Tgel value, compared to gelatin and GelMA from porcine origin that only showed a decrease of <1.3 °C (Table 3).

### 2.6. Thermal Characterization by DSC

The thermal property data obtained for gelatin and GelMA samples upon cooling at 10 and 20% *w/v* are summarized in Table 4 and Appendix A. These data suggested higher onset gelation temperatures for PGel samples when compared to SGel, in agreement with viscoelasticity analysis (Table 3). The change in enthalpy (ΔH_gel_) in SGel associated with the gelation temperature showed a decreasing trend, although not significant, when the pH decreased from 4.8 to 3.6. Changes in enthalpy are directly proportional to the relative amount of triple helical structures present in the polymer [69], implying that a decrease in pH is associated with a decrease in triple helix content, correlating with the rheology and CD results. Additionally, as reported from the CD measurements, the ΔH values associated with gelation in the case of SGelMA at pH 3.6 show a tendency to a lower content of triple helix formation than SGel. Regarding gelation temperatures for SGel and SGelMA, these are consistent with the rheological measurements, where the decrease in pH was found to significantly decrease the gelation temperature. The effect of pH on PGel/PGelMA and thermal transitions show a similar trend, with CD spectra and rheology data suggesting a slight decrease in the triple helix formation at lower pHs when cooling.

## 3. Discussion

In this work we evaluated the effect of pH and methacrylation on the molecular configuration and gelation properties of salmon (and porcine) gelatin. Methacrylation was found to decrease the IEP of derivatized gelatin (GelMA), probably due to a decrease in the number of positive charges along the gelatin’s backbone structure, and a decrease in the molecular weight of GelMA, as suggested by electrophoresis, capillary viscometry and MALDI-TOF. It is important to highlight that the molecular weight data that we provide are related to GelMA produced under the described conditions, since several methacrylation conditions can be found in the literature. These include differences in the type of buffer used (PBS pH 7.4 or carbonate buffer pH 9), the pH during the methacryloyl reaction given that some methods can involve pH control (others do not), temperature ranges (40–60 °C) and time periods (1–5 h) [13,17,20,21,25,59,70]. All these parameters can eventually affect the molecular weight of GelMA and the resulting viscoelastic and mechanical properties of the crosslinked hydrogels. Using the protocol described in our work (3 h, 60 °C), a decrease in Mw was expected since acidic pH conditions (due to the production of methacrylic acid as a byproduct) and a temperature of 60°C can favor gelatin hydrolysis [8]. This decrease in molecular weight was found to significantly affect viscoelastic and thermal properties. Other possible factors including lower interchain interaction due to amino acid chemical functionalization on gelatin polymer chains, however, may also have contributed to these property changes.

Additionally, pH can also affect the molecular structure of gelatin and GelMA, and thus their rheological and thermal properties, where we identified a decrease in triple helix formation and gelation temperatures when the pH was decreased from 4.8 to 3.6. With CD measurements, these changes in molecular structure were more evident in the case of porcine gelatin, since in the case of salmon-origin gelatin samples, triple helix formation was not complete. Interestingly, rheology analysis showed that the SGel and SGelMA molecular structures were more sensitive to the pH, showing significant changes in Tgel and triple helix formation, though porcine gelatin molecular structure modulation by pH was less evident. This is not as clear with DSC measurements, where more error is associated with ΔH_gel_ and T_onset_ measurements; thus, rheology seems to be the most sensitive technique for evaluating molecular structure changes in relation to low-viscosity gelatin samples.

Even with a high degree of functionalization, PGelMA still showed thermal instability at room temperature (gelation point). Hence, SGelMA can be regarded as a more suitable alternative material for complex applications, since tight temperature control would not be necessary, offering a simpler production setting. However, our results suggest that pH must be controlled tightly. These data agree with previous studies that suggested that SGelMA presented a higher molecular mobility than mammalian-origin GelMA [36]. This higher molecular mobility could affect triple helix stabilization at a specific temperature and pH, and therefore produce a decrease in triple helix formation and Tgel (as we observed), contributing to a higher thermal stability. This higher molecular mobility, as well as differences in their aminoacidic content, could also contribute to the higher pH sensibility of SGel and SGelMA in comparison to porcine samples.

Our results also suggest that methacrylation and pH control provide additional ways to control the viscosity, viscoelastic and thermal properties of salmon gelatin suspensions, which could be highly relevant for food and biomedical engineering applications, specifically for the development of bioinks and controlled release systems, where tight control of the final hydrogel structure formed is extremely relevant for their applications. These findings highlight the importance of a proper GelMA molecular configuration characterization prior to hydrogel fabrication, since small temperature and pH changes can affect gelatin and GelMA molecular configuration, thus impacting the final hydrogel structure after chemical crosslinking.

## 4. Materials and Methods

### 4.1. Reagents

Porcine gelatin type A Bloom 300 (cat. G2500) was purchased from Sigma Aldrich (St. Louis, MO, USA). Reagents used for salmon gelatin extraction, such as sodium hydroxide (NaOH) (cat. 106498) and glacial acetic acid (cat. 10063), were purchased from Merck (Darmstadt, Germany). Most of the other reagents used were purchased from Sigma Aldrich (St. Louis, MO, USA), including methacrylic anhydride (cat. 276685), O-phtalaldehyde (cat. P1378), di-sodium-tetraborate decahydrate (cat. S9640), dithiothreitol (DTT) (cat. D9779), and L-serine (cat. S4500), or at Merck, including Phosphate Buffered Saline (PBS) 10X (cat. 6505), sodium dodecyl sulfate (SDS) (cat. 428015) and absolute ethanol (cat. 100983).

### 4.2. Gelatin Production from Salmon Skin

Salmon gelatin was extracted from Atlantic salmon (*Salmo salar*) skins following the protocol reported by Diaz-Calderon et al., 2017 [8] with some modifications. Fish muscles and scales were removed from salmon skins using a sharp knife. Clean skins were rinsed with water and subsequently cut into squared pieces with surface areas of ~2.5 cm^2^. These were subsequently drained and stored at −20 °C until further use. The skins were subjected to a three-step pre-treatment before performing gelatin extraction. First, the skins were incubated in NaOH 0.1 M with constant agitation for 1 h at 10 °C. The solution was discarded, and the skins were rinsed with cold running water. This process was repeated once again under the same conditions. After rinsing, the skins were subjected to an acid pre-treatment in acetic acid 0.05 M, with constant agitation for 1 h at 10 °C. The solution was discarded, and the skins were rinsed with cold running water. Gelatin was extracted by incubating the pre-treated skins with distilled water at pH 4 (adjusted with acetic acid) under constant agitation for 3.5–4 h at 60 °C, checking and adjusting at pH 4 every hour. After the extraction step, the suspended solids were discarded and the resulting gelatin solution was vacuum filtered (22 µm pore size, cat. 1541–090, Whatman, UK). The filtered solution was then poured onto Teflon trays and dried at 60 °C for ~36 h in a forced convention oven (WiseVen WOF 105, Wonju, Kangwon-do, Republic of Korea).

### 4.3. Glycine, Proline and Hydroxyproline Content

The amino acid concentration of gelatin samples was determined by reverse-phase high-performance liquid chromatography (HPLC-RP) as previously reported [71], with modifications [8]. A liquid chromatograph (Waters 600 controller, Milford, MA, USA) with a diode array detector (Waters 996) and a Luna RP18 column (150 mm × 4.6 mm, particle size 5 mm) was used. Amino acid quantification was carried out using external standards of each analyzed amino acid (Sigma-Aldrich, Steinheim, Germany). The glycine, proline, hydroxyproline, and lysine content of salmon and porcine gelatin samples were reported as mmol/g of gelatin and as percentage (%) of total amino acids present in the sample.

### 4.4. Gelatin Functionalization

Salmon and porcine gelatin was functionalized with methacryloyl groups following the protocol proposed by van den Bulcke et al., 2000 [15] with modifications. A 10% *w/v* gelatin suspension was prepared in PBS 1 X, stirred for 1 h at 60 °C and then functionalized by adding methacrylic anhydride dropwise, to achieve an 8% *v/v* concentration. The reaction was conducted for 3 h, at pH 4 and 60 °C under constant stirring. The reaction was terminated by the addition of 3-fold PBS 1x by volume. The suspension was then dialyzed (dialysis tubing cat. D9402, Sigma Aldrich, St. Louis, MO, USA) for 4 days in distilled water, with two water changes per day until a suspension conductivity of <100 µS was achieved. The suspension was vacuum filtered (8 µm pore size, cat. 1440–090, Whatman, UK), freeze-dried (FDU-7020, Operon Co., Ltd., Gyeonggi-do, Republic of Korea), stored at −40 °C, and protected from light until further use.

### 4.5. Determination of Degree of Functionalization in GelMA

To corroborate the GelMA degree of functionalization, two methods were used: a colorimetric method using O-Phtalaldehyde (OPA) and Proton Nuclear Magnetic Resonance (^1^H-NMR) spectroscopy.

#### 4.5.1. O-Phtalaldehyde Method

The degree of functionalization of GelMA samples was determined using the OPA method [36,72]. For 200 mL of OPA reagent, 7.6 g of di-Na-tetraborate decahydrate and 200 mg of SDS were dissolved in 150 mL of distilled water (solution A). Then, 160 mg of OPA was dissolved in 4 mL of absolute ethanol (solution B). Once both solutions were completely dissolved, solution B was transferred to solution A, 176 mg of DTT was added, and the solution was made up to 200 mL with distilled water [72]. Samples of SGel, SGelMA, PGel and PGelMA at 10 mg/100 mL were prepared in distilled water and then diluted as necessary. Then, 200 µL of each sample was added with 1.5 mL of OPA reagent and incubated for 2 min before measurement. The absorbance of the samples was measured using a spectrophotometer at 340 nm (UV-1800, Rayleigh, Beijing, China). A standard curve for primary amine concentration with serine was used, and distilled water was used as blank. The percentage of the functionalization of amine groups in SGelMA and PGelMA samples was calculated in respect to an average value of amine groups in SGel and PGel, respectively.

#### 4.5.2. Proton Nuclear Magnetic Resonance (^1^H-NMR) Spectroscopy

The degree of functionalization was also determined using ^1^H-NMR with a Bruker 800 MHz Avance III spectrometer (Billerica, MA, USA) equipped with a QCI cryoprobe. Gelatin and GelMA samples were prepared in D_2_O at 50 mg/mL, and 600 µL of each sample was added to NMR tubes. All measurements were performed at 25 °C. The 1D ^1^H-NMR spectra were recorded using a 1D NOESY sequence with a spectral width of 14 ppm using on-resonance pre-saturation for water suppression. The proton transmitter frequency was set to 4.702 ppm and typically 64 scans were acquired. Data acquisition and processing were carried out by using the Topspin 3.1 software (Brukers, Billerica, MA, USA). For the quantification of the degree of functionalization, we used the method described by Hoch et al., 2012 [18]. The spectra were normalized to the aromatic protons from the phenylalanine signal (6.9–7.5 ppm) and the lysine methylene signals (2.95–3.00 ppm) of gelatin, and the GelMA spectra were integrated. The degree of functionalization (DF) of PGelMA and SGelMA samples was calculated as:(1)$$\mathrm{DF}\;\%\;=\left(1-\frac{\mathrm{lysine}\;\mathrm{methylene}\;\mathrm{peak}\;\mathrm{area}\;\mathrm{from}\;\mathrm{GelMA}\;}{\mathrm{lysine}\;\mathrm{methylene}\;\mathrm{peak}\;\mathrm{area}\;\mathrm{from}\;\mathrm{gelatin}}\right)\times 100$$

### 4.6. Determination of Molecular Weight

#### 4.6.1. SDS-PAGE

Gelatin molecular weight distribution was assessed by SDS-PAGE electrophoresis, using 7.5% acrylamide pre-casted gels (7.5% Mini-Protean^®^ TGX™ precast protein, 10 well, BioRad, Hercules, CA, USA). Samples were heated at 95 °C for 5 min before loading (50 µg of gelatin and GelMA in distilled water), and standard molecular weight markers in the 250–10 KDa range were used (Kaleidoscope™, Precision Plus Protein Standards™, BioRad, Hercules, CA, USA). The electrophoresis was run at 100 V for 40 min and the resulting gel was stained with Bio-Safe Coomassie Brilliant blue G-250 stain (BioRad, Hercules, CA, USA) for 30 min, following the supplier’s instructions.

#### 4.6.2. Capillary Viscometry

Capillary viscometry was used to determine an average molecular weight for porcine and salmon gelatin samples. The dependence of the reduced viscosity (η_red_) and inherent viscosity (η_inh_) of a dilute polymer suspension on the concentration (c) is well established [73]. The effect of the dispersion of a macromolecule in a solution is given by the relative viscosity (η_rel_) or (η_red_) as follows:(2)$$\eta_{\mathrm{rel}}=\frac{\eta }{\eta_{0}}$$
(3)$$\eta_{\mathrm{red}\;}=\left(\frac{\eta_{\mathrm{rel}}-1}{c}\right)$$
where η is the viscosity of the dispersion, η_0_ is the viscosity of the solvent [61]. On the other hand, η_inh_ is defined as: (4)$$\eta_{\mathrm{inh}}=\left(\frac{{\ln\;\eta }_{\mathrm{rel}}}{c}\right)$$

At infinite dilution (c→0), η_red_ and η_inh_ are defined as the intrinsic viscosity [η] [73]. The relation between [η] and average molecular weight (Mw) can be determined with the following empirical Mark–Houwink Kuhn–Sakurada (MHKS) equation:(5)$$\left[\eta \right]={K\times \mathrm{Mw}}^{a}$$
where K and a are constants that are dependent on the nature of the solvent and the polymer conformation [61]. The determination of Mw for each gelatin sample was carried out using the values of a and K used by Veis, 1964 [74] using the following equation:(6)$$\left[\eta \right]=8.6\times 10^{-5}\cdot {\mathrm{Mw}}^{0.74}$$

Measurements were performed as previously described [55]. Briefly, concentrations from 2 to 6 g/L of each sample were prepared in 0.1 M NaCl and were left overnight at 4 °C. The flow time of each concentration was determined by measuring the time required for the suspension to flow from the top to the bottom mark of the viscometer (size 50, Z275271-1EA, Sigma Aldrich, St. Louis, MO, USA). Each flow time was measured four times, and the experiment was performed twice for each sample. η_red_ and η_inh_ values were plotted against concentration, and the point of convergence between η_red_ and η_inh_ values was taken as the intrinsic viscosity [η].

#### 4.6.3. MALDI-TOF Mass Spectroscopy

To complement the SDS-PAGE and capillary viscometry analyses, SGel and SGelMA samples were analyzed by matrix-assisted laser desorption/ionization time-of flight mass spectroscopy (MALDI-TOF MS) as described before by Liu et al., 2012 [75]. Samples were mixed with 5 µL of trifluoroacetic acid (TFA) 0.1% *v/v* and 5 µL of matrix solution (10 mg/mL of synaptic acid in a 50:50 acetonitrile mixture). Then, 1 µL of this mixture was applied to a target plate. Mass spectra were obtained using MALDI-TOF MS Autoflex Speed (Bruker Daltonics, Bremen, Germany) equipped with a smart beam (334 nm) source. Spectra were obtained using a positive and lineal mode with an accelerating voltage of 20 kV. Each spectrum was collected as an average of 1200 laser shots with sufficient energy to produce good spectra.

### 4.7. ζ-Potential and Isoelectric Point

The ζ-potential was determined with electrophoretic light scattering using a Malvern Zetasizer (Nano ZS, Malvern Instruments, Malvern, UK) with disposable folded capillary cells (DTS1070). Prior to the measurement, all gelatin and GelMA suspensions were prepared at 1.5 g/L concentrations for suitable scattering intensity. Measurements were acquired at a 4–11 pH range, every 0.5 pH value. The isoelectric point (IEP) was determined as the pH value for which the corroborated potential was 0 mV.

### 4.8. Secondary Structure Molecular Configuration

The molecular configuration of each gelatin and GelMA sample was studied by circular dichroism (CD) (Chirascan Plus Spectrometer, Applied Photophysics, Leatherhead, UK). Porcine gelatin (PGel) with its well-defined triple helix formation at room temperature was used as a standard. All gelatin and GelMA suspensions were prepared at 0.1% *w/v* concentration. At least three spectral scans were performed on each sample at wavelengths ranging from 180 to 260 nm with 0.5 nm increments and with a dwell time of 0.5 s. A baseline scan of distilled water was also run. Solutions were pipetted into a quartz cuvette cell of path-length 100 mm, and control spectra were generated for each sample after heating to 40 °C and holding isothermally for 120 min to ensure the helix-to-coil transition occurred for all the samples. The gelatin suspensions were then placed into a cold room at 4 °C for 120 min and then held in the spectrometer at 4 °C for 5 min to allow thermal equilibrium prior to the measurements. The final data were averaged over the three scans and the baseline spectrum was subtracted.

### 4.9. Rheological Measurements

All gelatin and GelMA samples were prepared at 10% *w/v* by adding 1 g of sample powder to 10 mL of distilled water. The pH of each suspension was measured at 55 °C using a standard pH meter. Hydrochloric acid (HCl) 5 M and/or sodium hydroxide (NaOH) 5 M was added to the suspensions to adjust the pH to 4.8 or 3.6. It is important to state that these values are related to the reduction in the pH of the gelatin due to the addition of methacrylic anhydride during functionalization. This was performed in an effort to determine if differences in gelatin and GelMA molecular structure were due to molecular changes or differences in pH values. All rheological measurements were performed using a rheometer Discovery HR-2 (TA Instruments, New Castle, DE, USA), using 5 cm parallel plate geometry (PLATE SST ST 5 CM), a 300 µm gap, and a solvent trap to avoid water evaporation during the analysis. Viscosity measurements were performed during a cooling temperature ramp from 40 to −5 °C at a cooling rate of 3 °C/min using a shear rate of 1600 s^−1^. Steady-state shear viscosity at 20 °C was also performed, where shear rate values between 10 and 1500 s^−1^ were applied. All measurements were performed at least in duplicate. 

To determine viscoelastic properties, storage modulus (G′) and loss modulus (G″) curves were measured upon cooling from 40 to −5 °C at 3 °C/min. A 10% deformation and 1 Hz frequency were applied, which was within the linear region of the stress–strain curve as previously determined. The gelation temperature was defined when G′ and G″ curves crossed over upon the cooling ramp. Each measurement was performed in duplicate.

### 4.10. Thermal Properties

The thermal properties of gelatin and GelMA suspensions, namely the gelation temperature (T_gel_) and changes in gelation enthalpy (ΔH_gel_), were assessed by differential scanning calorimetry (DSC) (DSC 1 STAR System, Mettler-Toledo, Greinfensee, Switzerland) using an intracooler TC100 (HUBER, Raleigh, NC, USA). The measurements were carried out by loading ~70 mg of each suspension (10% and 20% *w/v*) into a stainless-steel pan (120 µL). An empty pan was used as a reference and gas N_2_ was used as purge gas. The thermal cycle included cooling from 20 to −15 °C at 3 °C/min, an isotherm step at −15 °C for 5 min, heating up to 80 °C at 10 °C/min and an isotherm step at 80 °C for 5 min. The samples were subjected to the same thermal protocol twice to erase the material’s thermal history. T_gel_ and ∆H were determined from cooling scans using STARe Software (DB V12.10, Mettler-Toledo, Greinfensee, Switzerland). Prior to the measurements, the temperature and enthalpy were calibrated at a heating rate of 10 °C/min using indium as standard (Tm = 156.6 °C and ΔHm = 28.55 J/g).

## 5. Conclusions

During this work we evaluated the effect of pH and methacrylation on the molecular configuration and gelation properties of gelatins of salmon and porcine origin, where we determined that methacrylation decreased the molecular weight of the derivatized gelatins. Additionally, pH also affected the molecular structure of gelatin and GelMA. We identified a decrease in triple helix formation and gelation temperatures when the pH was decreased from 4.8 to 3.6. This decrease was more evident in salmon origin samples than gelatin samples of porcine origin. These findings highlight the importance of a proper GelMA molecular configuration characterization prior to hydrogel fabrication, since small temperature and pH changes can affect gelatin and GelMA molecular configuration and thus the final hydrogel structure after chemical crosslinking. Our results also suggest that methacrylation and pH control provide additional ways to control the viscosity, viscoelastic and thermal properties of salmon gelatin suspensions, highly relevant for biomedical engineering applications such as biofabrication.

## Figures and Tables

**Figure 1 ijms-24-07489-f001:**
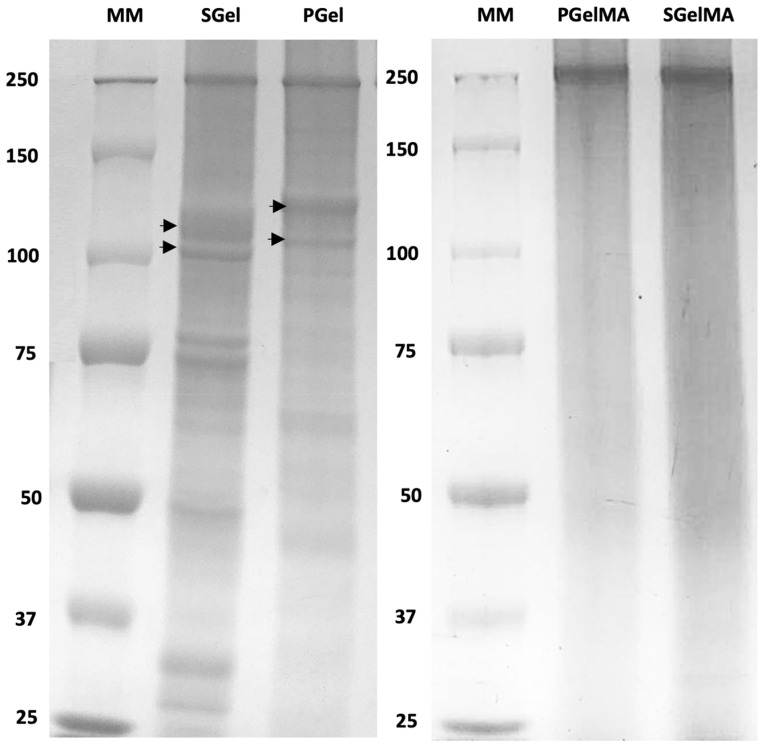
SDS-PAGE of gelatin samples. MM: corresponds to the molecular weight marker in kDa. Arrowheads indicate α-chains present in gelatin from salmon (SGel) and porcine origin (PGel).

**Figure 2 ijms-24-07489-f002:**
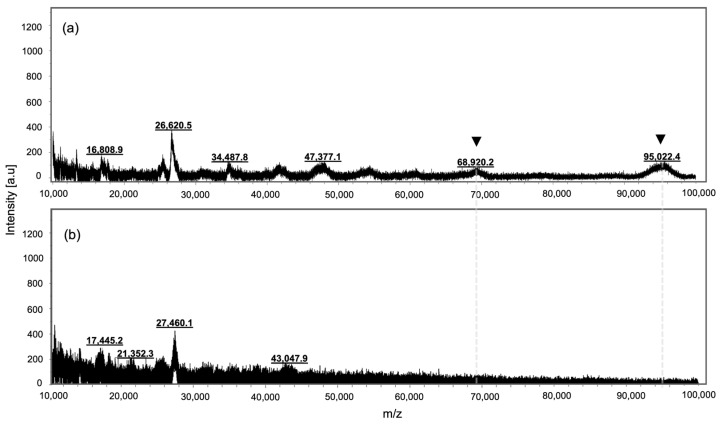
MALDI-TOF mass spectra for salmon gelatin (**a**) and GelMA (**b**) samples. Arrowheads indicate Mw bands from SGel absent from SGelMA samples.

**Figure 3 ijms-24-07489-f003:**
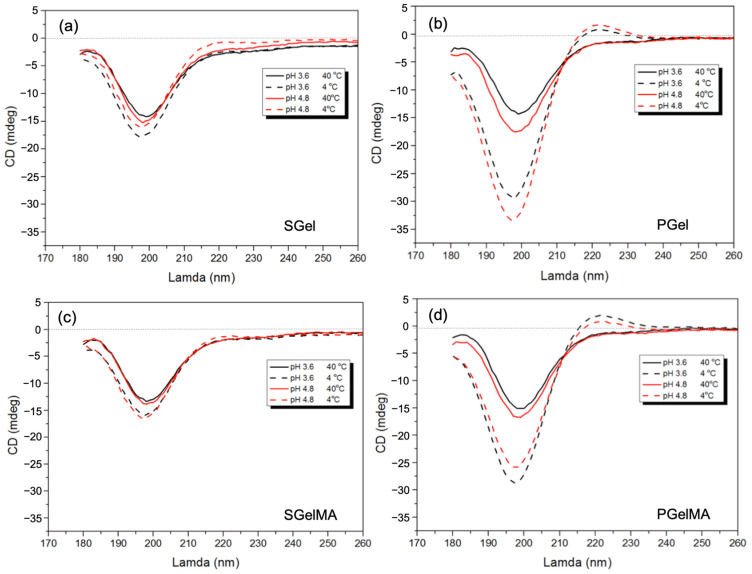
CD spectra assessed at temperatures above T_gel_ (40 °C) and below T_gel_ (4 °C) of SGel (**a**) and PGel (**b**) both at pH ~3.6 and 4.8 and their methacryloyl counterparts SGelMA (**c**) and PGelMA (**d**) both at pH ~3.6 and 4.8.

**Figure 4 ijms-24-07489-f004:**
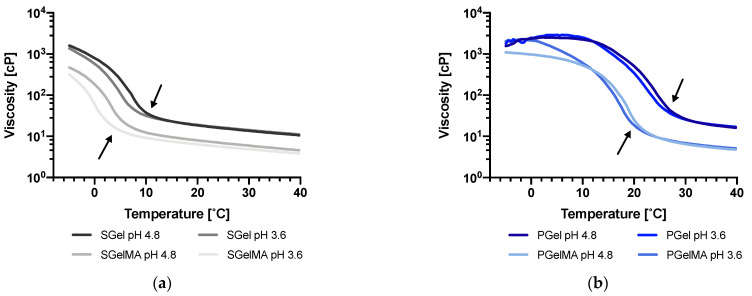
Flow temperature ramp (cooling) of gelatin suspensions (10% *w*/*v*) at pH ~3.6 and 4.8. (**a**) SGel and SGelMA, (**b**) PGel and PGelMA. Arrows indicate the marked increase in viscosity for all suspensions.

**Figure 5 ijms-24-07489-f005:**
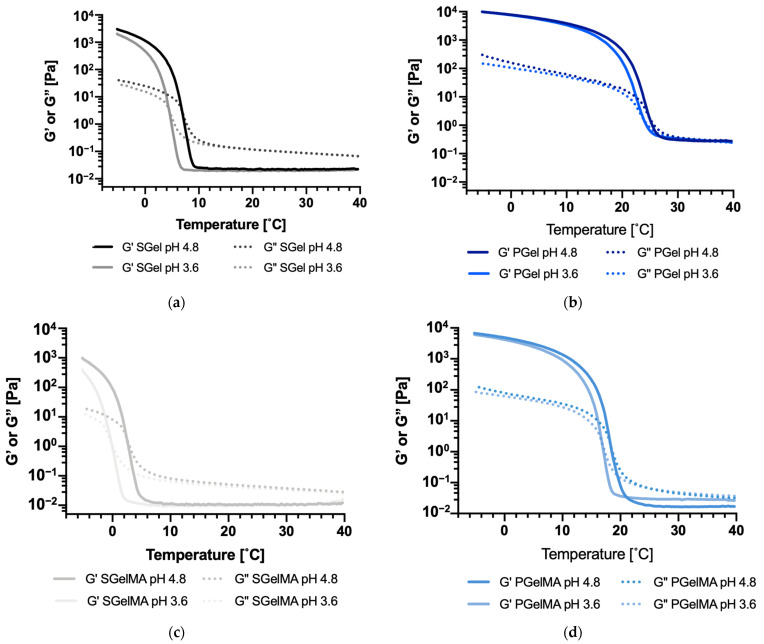
Storage (G′) and loss modulus (G″) for gelatin and GelMA suspensions (10% *w*/*v*) at pH ~3.6 and ~4.8 at different temperatures (cooling ramp from 40 to −5 °C). (**a**) SGel, (**b**) PGel, (**c**) SGelMA, (**d**) PGelMA.

**Table 1 ijms-24-07489-t001:** Glycine, proline, hydroxyproline, and lysine contents of salmon and porcine gelatins. Values are presented as average ± standard deviation.

Amino Acid	Content [mmol/g]		% From Total Amino Acids	
	SGel	PGel	SGel	PGel
Glycine	2.40 ± 0.02	2.60 ± 0.03	25.07 ± 0.24	28.33 ± 0.18
Proline	0.57 ± 0.01	0.88 ± 0.01	9.08 ± 0.01	14.79 ± 0.01
Hydroxyproline	0.45 ± 0.02	0.76 ± 0.01	8.19 ± 0.27	14.62 ± 0.19
Lysine	0.19 ± 0.01	0.18 ± 0.01	3.78 ± 0.06	3.95 ± 0.09

**Table 2 ijms-24-07489-t002:** Degree of functionalization (DF), Intrinsic viscosity [η] and average molecular weight (Mw) values determined by capillary viscometry, and isoelectric point (IEP) values for different gelatin and GelMA samples. Values are presented as average ± standard deviation.

Sample	DF [%] by OPA	DF [%] by ^1^H-NMR	[η] [mL/g]	M_w_ [kDa]	IEP at 25 °C [mV]
SGel	-	-	34.17 ± 0.25	73.40 ± 0.73	9.2
PGel	-	-	38.70 ± 0.69	86.87 ± 2.09	9.4
SGelMA	96.3 ± 0.3	91.9	14.61 ± 0.13	23.28 ± 0.27	4.9
PGelMA	96.5 ± 0.1	83.2	13.44 ± 0.03	20.81 ± 0.06	4.8

**Table 3 ijms-24-07489-t003:** Gelation temperature (Tgel) of different gelatin suspensions (10% *w*/*v*) determined by rheology, and differences in Tgel due to differences in pH (ΔTgel). Values are presented as average ± standard deviation.

Sample	Tgel pH 4.8 [°C]	Tgel pH 3.6 [°C]	ΔTgel [°C]
SGel	7.0 ± 0.2	4.5 ± 0.2	2.5
PGel	25.3 ± 0.1	24.2 ± 0.2	1.1
SGelMA	2.4 ± 0.3	−0.4 ± 0.6	2.8
PGelMA	18.5 ± 0.3	17.2 ± 0.2	1.3

**Table 4 ijms-24-07489-t004:** Enthalpy (ΔH_gel_) and gelation temperatures (T_onset_) obtained for SGel and SGelMA upon cooling. Values are presented as average ± standard deviation.

Sample	10% *w*/*v*		20% *w*/*v*	
	Normalized ΔH_gel_ [J/g]	T_onset_ [°C]	Normalized ΔH_gel_ [J/g]	T_onset_ [°C]
SGel pH 4.8	4.9 ± 0.9	10.2 ± 0.5	6.5 ± 0.3	10.6 ± 0.1
SGel pH 3.6	2.9 ± 2.0	7.2 ± 0.7	5.7 ± 1.1	7.6 ± 0.4
SGelMA pH 3.6	2.5 ± 0.4	7.0 ± 0.6	3.7 ± 0.7	8.6 ± 2.2
PGel pH 4.8	2.8 ± 1.1	22.4 ± 1.4	9.2 ± 4.1	27.3 ± 1.4
PGel pH 3.6	3.9 ± 0.6	20.8 ± 1.8	6.4 ± 0.6	22.6 ± 0.4
PGelMA pH 3.6	2.0 ± 0.1	18.8 ± 1.3	2.7 ± 0.9	20.8 ± 1.5

## Data Availability

The data presented in this study are available on request from the corresponding author.

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
