# Peer review of "Understanding the Molecular Conformation and Viscoelasticity of Low Sol-Gel Transition Temperature Gelatin Methacryloyl Suspensions"

_ijms, 2023, doi:10.3390/ijms24087489_

Round 1
Reviewer 1 Report
Applications of this research and current need should be focused somewhere in introduction.
Also, there is no Conlcusion section, it must be added.
Author Response
Dear Reviewer,
We thank you for the recommendations and suggestions on our manuscript. We have revised and corrected it accordingly. The changes in the reviewed manuscript are marked up using the “Track Changes” tool. The following is our response to your specific comments:
- Applications of this research and current need should be focused somewhere in introduction.
We included more information in the introduction and modified previous information regarding GelMA hydrogels' benefits and drawbacks as well as their potential applications (lines 76 - 97), to highlight the importance and current need for this research. Also, we marked in yellow previous section of the manuscript that we believe covers as well the current need for this research (lines 139-151).
- Also, there is no Conclusion section, it must be added.
We included a conclusion section (lines 646 to 659 of the manuscript).
We thank you again for the detailed revision and the valuable comments on our manuscript.
Best Regards,
Prof. Javier Enrione
Universidad de los Andes.
Reviewer 2 Report
Comments and Suggestions to Authors
The manuscript titled “Understanding of molecular conformation and viscoelasticity of low
sol-gel transition temperature gelatin methacryloyl suspensions” is of readers’ interest and within
the journal’s scope. However, the manuscript needs some revision. The concerns and/or
suggestions along with some questions are listed below for the author’s attention.
1. How do the molecular configurations of salmon gelatin (SGel) and salmon methacryloyl
gelatin (SGelMA) at two different acidic pHs (3.6 and 4.8) compare to those of commercial
porcine gelatin (PGel) and methacryloyl porcine gelatin (PGelMA)?
2. How does functionalization with methacryloyl groups affect the molecular weight and
isoelectric point (IEP) of the gelatins?
3. What are the effects of functionalization and pH on the molecular structure, rheological,
and thermal properties of the gelatins?
4. How does the sensitivity of SGel and SGelMA molecular structure to pH changes compare
to that of PGel and PGelMA, specifically regarding gelation temperatures and triple helix
formation?
5. Can SGelMA be considered a highly tunable biomaterial for biofabrication applications,
and how important is the proper characterization of GelMA molecular configuration before
hydrogel fabrication?
6. Introduction: Please provide a more comprehensive review of the existing literature on
gelatin methacryloyl (GelMA), including its advantages and limitations. Additionally,
discuss the importance of molecular conformation and viscoelasticity for biomedical
applications and biofabrication.
7. Materials and Methods: Provide more details on the preparation and characterization of the
gelatin samples, including the specific methods used for molecular weight determination,
isoelectric point measurement, circular dichroism, and rheological and thermophysical
property analysis. This will help the reader better understand the methodology and ensure
reproducibility.
8. Results: Please present the results in a more organized and structured manner, possibly
using tables or figures to help readers easily compare the data. Make sure to clearly describe
the differences and similarities between the various gelatin samples and their molecular
conformations, rheological properties, and thermophysical properties at different pH
levels.
9. Discussion: Expand the discussion section to provide more in-depth analysis of the results
obtained. Specifically, discuss the impact of functionalization and pH on the molecular
structure, rheological, and thermal properties of the gelatins. Also, explain why the salmon-
derived gelatins showed greater sensitivity to pH changes compared to mammalian-derived
gelatins.
10. Conclusion: Summarize the main findings of the study, including the significance of
understanding the molecular conformation and viscoelasticity of low sol-gel transition
temperature gelatin methacryloyl suspensions. Highlight the potential applications of
salmon-derived gelatin methacryloyl (SGelMA) in biofabrication and emphasize the
importance of proper characterization before hydrogel fabrication.
11. References: Ensure that all the references cited in the manuscript are up-to-date and
relevant to the study. Double-check the formatting and consistency of the reference list
according to the journal’s guidelines.
12. Language and Clarity: Review the manuscript for grammar, punctuation, and sentence
structure errors. Make sure that the language is clear and concise throughout the
manuscript, and that the scientific terms are defined and used consistently.
Addressing these concerns and suggestions, as well as answering the above questions will
improve the quality and clarity of this manuscript, making it more accessible and informative
for the readers.
Author Response
Dear Reviewer,
We thank you for the recommendations and suggestions on our manuscript. We have revised and corrected it accordingly. The changes in the reviewed manuscript are marked up using the “Track Changes” tool. The following is our response to your specific comments:
- How do the molecular configurat ions of salmon gelatin (SGel) and salmon methacryloyl
gelatin (SGelMA) at two different acidic pHs (3.6 and 4.8) compared to those of commercial
porcine gelatin (PGel) and methacryloyl porcine gelatin (PGelMA)?
The molecular configuration of SGel and SGelMA at two different pHs are different from those of commercial PGel and PGelMA. We marked in yellow the observed differences in the circular dichroism section (lines 269-275 & 306-310), rheological characterization section (lines 308-330 & 338-343), and we included a small summary of the main findings before introducing the thermal characterization data (lines 353-358). We also modified the discussion to highlight the main changes observed between the molecular configuration of SGel and SGelMA from those of commercial PGel and PGelMA (lines 416-422).
- How does functionalization with methacryloyl groups affect the molecular weight and
isoelectric point (IEP) of the gelatins?
Functionalization with methacryloyl groups decreases the molecular weight and
isoelectric point of gelatins from salmon and porcine origin as evidenced in section 2.3. We modified SDS-PAGE and Maldi-TOF images to better explain these differences in a visual manner. The possible explanations associated with IEP changes are marked in yellow (lines 257-262) in the results section and in the discussion section we marked in yellow an explanation for the decrease in IEP and molecular weight (lines 249-254).
- What are the effects of functionalization and pH on the molecular structure, rheological,
and thermal properties of the gelatins?
This is in relation to the first and second questions since the rheological and thermal properties of GelMA and gelatin are a consequence of their molecular configuration, which is affected by a decrease in the pH (question 1). The main effect of functionalization on gelatin molecular structure is a decrease in the molecular weight and isoelectric point of gelatins (question 2). Also, the functionalized gelatins (GelMAs) molecular configuration, can also be affected by a decrease in pH, mainly evidenced by rheological measurements (question 1).
- How does the sensitivity of SGel and SGelMA molecular structure to pH changes compare
to that of PGel and PGelMA, specifically regarding gelation temperatures and triple helix
formation?
Differences in pH sensitivity, where SGel and SGelMA present higher pH sensitivity than PGel and PGelMA were commented on the rheological characterization section (lines 338-343), as well as in the small summary of the main findings before introducing the thermal characterization data (lines 353-358).
- Can SGelMA be considered a highly tunable biomaterial for biofabrication applications,
and how important is the proper characterization of GelMA molecular configuration before hydrogel fabrication?
This asseveration is further discussed in the discussion section and conclusion section, to highlight the importance of the information reported in this manuscript (lines 415-430) in discussion and lines 630-642 (in conclusion).
- Introduction: Please provide a more comprehensive review of the existing literature on
gelatin methacryloyl (GelMA), including its advantages and limitations. Additionally,
discuss the importance of molecular conformation and viscoelasticity for biomedical applications
We included more information in the introduction and modified previous information regarding GelMA hydrogels' benefits and drawbacks as well as their potential applications (lines 76 to 96), to highlight the importance and current need for this research. Also, we marked in yellow the previous section of the manuscript that we believe covers as well the current need for this research (lines 139-351).
- Materials and Methods: Provide more details on the preparation and characterization of the gelatin samples, including the specific methods used for molecular weight determination, isoelectric point measurement, circular dichroism, and rheological and thermophysical property analysis. This will help the reader better understand the methodology and ensure
We tried to include more information in the different methodologies, however, we believe the methods descriptions are very clear and detailed enough to ensure reproducibility.
- Results: Please present the results in a more organized and structured manner, possibly
using tables or figures to help readers easily compare the data. Make sure to clearly describe the differences and similarities between the various gelatin samples and their molecular conformations, rheological properties, and thermophysical properties at different pH
We modified SDS-PAGE and Maldi-TOF images to better explain the observed differences in molecular weight, however, aside from that we believe that our results are presented in an organized manner. We also modified different sections of the manuscript to try to explain our findings better.
- Discussion: Expand the discussion section to provide a more in-depth analysis of the results Specifically, discuss the impact of functionalization and pH on the molecular structure, rheological, and thermal properties of the gelatins. Also, explain why the salmon derived gelatins showed greater sensitivity to pH changes compared to mammalian-derived gelatins.
We expanded our discussion to further analyze our main findings, specifically regarding the main differences in the molecular configuration of the different samples as well as differences in pH sensitivity, and possible explanations for these differences were added. Also, we further highlighted the importance of the study for the development of different biomedical applications (lines 416-432).
- Conclusion: Summarize the main findings of the study, including the significance of
understanding the molecular conformation and viscoelasticity of low sol-gel transition
temperature gelatin methacryloyl suspensions. Highlight the potential applications of
salmon-derived gelatin methacryloyl (SGelMA) in biofabrication and emphasize the importance of proper characterization before hydrogel fabrication.
We included a conclusion section (lines 629-642) of the manuscript) that included your recommendations.
- References: Ensure that all the references cited in the manuscript are up-to-date and
relevant to the study. Double-check the formatting and consistency of the reference list
according to the journal’s guidelines.
We checked the format of the references from the existing and new references that were added.
- Language and Clarity: Review the manuscript for grammar, punctuation, and sentence
structure errors. Make sure that the language is clear and concise throughout the
manuscript, and that the scientific terms are defined and used consistently.
We checked and corrected several grammar and punctuation errors that we found.
We thank you again for the detailed revision and the valuable comments on our manuscript. We are convinced that they helped to improve the quality and clarity of the manuscript.
Best Regards,
Prof. Javier Enrione
Universidad de los Andes.